# Mistake-driven Image Classification with FastGAN and SpinalNet

## Abstract

Image classification with classes of varying difficulty can cause performance disparity in deep learning models and reduce the overall performance and reliability of the predictions. In this paper, we address the problem of imbalanced performance in image classification, where the trained model has performance deficits in some of the dataset's classes. By employing Generative Adversarial Networks (GANs) to augment these deficit classes, we finetune the model towards a balanced performance among the different classes and an overall better performance on the whole dataset. Specifically, we combine a light-weight GAN method, FastGAN (Liu et al., 2021), for class-wise data augmentation with Progressive SpinalNet (Chopra, 2021) and Sharpness-Aware Minimization (SAM) (Foret et al., 2020) for training. Unlike earlier works, during training, our method focuses on those classes with lowest accuracy after the initial training phase. Only these classes are augmented to boost the accuracy, which leads to better performance. Due to the use of a light-weight GAN method, the GAN-based augmentation is viable and effective for mistake-driven training even for datasets with only few images per class, while simultaneously requiring less computation than other, more complex GAN methods. Our extensive experiments, including ablation studies on all key components, show competitive or better accuracy than the previous state-of-the-art on five datasets with different sizes and image resolutions.

## 1 Introduction

Supervised training of deep learning models, like the image classification models we consider in this paper, is the most efficient training approach, but also the most data-intensive. To mitigate this issue, alternative techniques are commonly applied, e.g., transfer learning from a pretrained model to a new domain (Ridnik et al., 2021) or data augmentation to expand the available dataset synthetically (Shorten & Khoshgoftaar, 2019). One particular data augmentation technique is to use Generative Adversarial Networks (GANs) (Goodfellow et al., 2020) that learn to produce new data that is similar to the distribution of the GAN's training data. GAN-based augmentation has been shown to be successful in aiding the training process (Bowles et al., 2018; Wang et al., 2018c; Tanaka & Aranha, 2019) and has found practical adoption in other research domains (Frid-Adar et al., 2018; Sasmal et al., 2020).

One challenge and limitation of GAN-based augmentation is either the high computational cost to augment each class of the dataset individually or the selection of the classes to augment that best support the model training. In this paper, we focus on a mistake-driven training procedure (see Figure 1) that utilizes GAN-based data augmentation only after the initial model training, albeit transfer learning or training from scratch, and augments only those classes with the lowest accuracy, where the model shows performance weaknesses. This keeps the effort for class-wise GAN training at a lower level than a full-sized data augmentation procedure but still contributes to the model improvement by focusing on the most significant deficits accuracy-wise. The mistake-driven training method is fully compatible with any other training regime as it does not interfere directly with the model training or finetuning step except by querying the model during evaluation and enhancing the dataset for finetuning.

Due to this explicit focus on model weaknesses as well as the careful selection of the components in the mistake-driven training pipeline, our method is computationally efficient and shows compet-

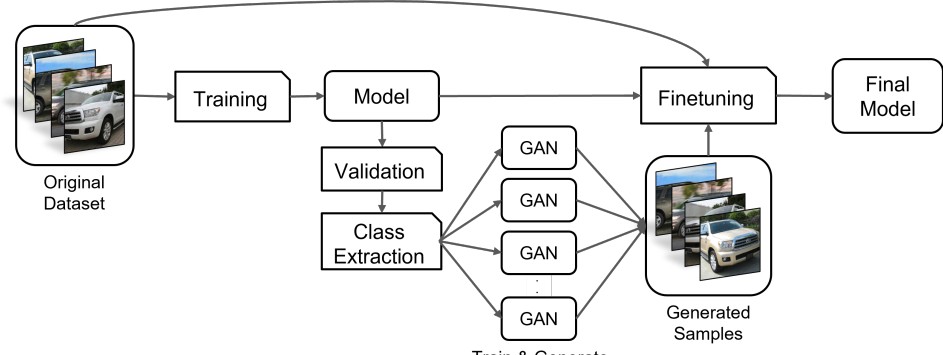

Figure 1: The workflow of the mistake-driven training process. After the initial training, the model is validated to identify the weakest classes, which are then expanded through GAN-based augmentation. The final finetuning step adjusts the performance imbalance between classes and boosts the overall performance.

itive performance on a number of benchmarks. First, we employ the progressive SpinalNet neural network architecture (Kabir et al., 2020; Chopra, 2021) as the image classification model, build on a backbone model, e.g. a pretrained Wide-ResNet (Zagoruyko & Komodakis, 2016) or EfficientNet-B7 (Tan & Le, 2019) as used in the experiments. Afterward, the class-wise accuracy on the validation set is computed, and the worst-performing classes are identified. For each of these classes, a sample-efficient GAN is trained, here we choose the light-weight FastGAN method (Liu et al., 2021), and use it to generate new samples. Finally, the already trained model is finetuned under consideration of the newly generated samples to improve and balance the model performance among the previously worst-performing classes.

We perform an extensive experimental evaluation to test the influence of each component in the mistake-driven training pipeline and our component selection of backbone models, activation functions, and model optimizers. The method is evaluated on five widely used image classification datasets and exceeds the previous state-of-the-art accuracy on four of them.

The contribution of our paper is threefold:

1. During training, our method focuses on those classes with the lowest accuracy after the initial training phase. Only these classes are augmented to boost the accuracy, which leads to better performance.

2. Due to the light-weight GAN method, the GAN-based augmentation is viable and effective for mistake-driven training even for datasets with only a few example images per class, while simultaneously requiring less computation than other, more complex GAN methods.

3. Our extensive experiments, including ablation studies, show competitive or better accuracy than the previous state-of-the-art on five datasets.

## 2 RELATED WORK

This section relates the presented work to the recent literature on data augmentation using GANs. There are two types of image augmentation methods that help in increasing the accuracy of the model and making the model more robust. The first one is a geometrical transformation-based augmentation, and the other one is generative adversarial-based augmentation.

**Geometrical Transformation & Data Augmentation** In the past decade, various data augmentation techniques have been used to improve classification prediction accuracy. The most common ones are those based on geometrical transformations (Shorten & Khoshgoftaar, 2019) e.g., cropping, flipping, rotation, color space, noise injection, and translation. These geometrical transformations increase the training dataset and improve the test accuracy (Perez & Wang, 2017). These data augmentation techniques are a best practice for training deep neural networks, especially in computer

vision tasks where a vast number of augmentations are available and can be combined. Cubuk et al. (2019) further introduces a method to perform this combination, called AutoAugment automatically, that learns dataset-specific augmentation strategies to increase the final model accuracy.

**Synthetic Data Generation Using Generative Models** Generative models, GANs in this context, have shown growing capability in generating very realistic data. These generated synthetic images can be utilized for data augmentation in data classification. Researchers have been attempting to enhance high-resolution images with GANs but with little success due to the enormous amount of data required to train GANs.

Data augmentation with GANs has recently received increasing attention from the research community. Some research works have attempted to supplement images using GANs for data classification (Tanaka & Aranha, 2019). However, with limited data, utilizing GANs to augment data becomes a strenuous process. Rashid et al. (2019), proposed to augment the ISIC skin lesion classification dataset using GANs. The results produced by data augmentation using GANs outperform ResNet and densenet models. In another line of work, Sasmal et al. (2020) used DCGAN for generating synthetic colonoscopic images. The generated images help in data augmentation to perform better polyp classification. Furthermore, Bejiga & Melgani (2018) suggests a GAN-based domain adaptation technique for aerial image classification. They use GANs for unsupervised domain adaptation of aerial remote sensing images. In the same field of work, Saha et al. (2021) developed TilGAN for the classification of images showing til, i.e., tumor-infiltrating lymphocytes and non-til images.

In addition to the previous research works, Zeng et al. (2020) propose to use GANs for augmenting data to detect disease severity. In another line of work, GANs were used for data augmentation in the field of multi-domain learning (Yamaguchi et al., 2020). These multi-domain GANs learn both the outer and target datasets simultaneously and generate new samples for the target tasks. A detailed survey on the use of GANs for data augmentation is presented by Shorten & Khoshgoftaar (2019). In this paper, our focus is to augment high-resolution images with a limited dataset which can further be used for finetuning the model. However, if the amount of the training data is not a constraining factor, IDA GAN (Yang & Zhou, 2021) and Polarity GAN (Deepshikha & Naman, 2020) can be utilized for data augmentation, too.

**Data Augmentation with Limited Data** GANs generally require massive datasets for training; limited data makes learning the underlying model arduous. Moreover, training GANs with limited data can easily lead to overfitting (Bowles et al., 2018), which makes the training more difficult. Transfer learning can be used to avoid the problem of overfitting of GANs, which also results in better performance (Bengio, 2012). Apart from the traditional way of augmenting data using geometric transformations, some GANs use observations that have been forged (Wang et al., 2018b) for generating new samples. Even though generating new images with limited data using GANs remains a challenging task, but at least the risk of overfitting can be reduced by using a pretrained model on a huge dataset and using that pretrained model for training GANs can produce some good quality images Wang et al. (2018b). However, using this technique sometimes leads to mode collapse, which is further fixed by Liu et al. (2021) using skip connections while training.

## 3 BACKGROUND

### 3.1 SHARPNESS-AWARE MINIMIZATION

In deep learning models, we need to optimize the loss function such that the DL model can converge to global minima. To reach the minima, many optimization algorithms such as stochastic gradient descent, Adam (Kingma & Ba, 2014), and RMSProp are being used as a design choice for optimizing the loss function. Zhang et al. (2016) show that DL models could memorize the training data and easily overfit to it, due to which the trained model lack generalization ability. However, when a DL model converges to an abrupt global minimum, the value of the loss function remains high in the neighborhood of the global minimum, causing the DL model to lose its generalization ability.

For the better generalization of DL models, Sharpness-Aware Minimization (SAM) was proposed (Foret et al., 2020), where the optimization goal is reformulated such that it considers the output of loss functions from not only the minimum but also at its neighboring points. In that way, SAM

minimizes both loss value and loss sharpness. Finally, the minimization of loss value and loss sharpness is done using gradient descent by learning those parameters that converge to a global minima, which has nearly the same loss value at the neighboring points.

## 3.2 PROGRESSIVE SPINAL NETWORKS

In the human biological network, the spine performs the preprocessing of the input, which then goes to the brain for the final output (D'Mello & Dickenson, 2008). By taking inspiration from the human spinal cord, Kabir et al. (2020) proposed spinal networks.

As the human brain receives information from various sensory neurons, and these signals are processed by the human spine before reaching the brain. Progressive SpinalNet consists of a processing unit and concatenating unit. The input to the Progressive SpinalNet Fully Connected (FC) layer is the CNN features. The processing unit can be a single hidden layer or multiple hidden layers. Each layer receives some portion from the input. The output of the processing unit is concatenated, and the following processing unit receives the concatenated output as an input. The gradients are easily propagated back to the first FC layer from the last FC layer due to connections between the layers. This helps in dealing with the vanishing gradient problem. The size of each layer is progressively increased, and the output size is the same for all the FC layers. The Progressive SpinalNet can be deeper as it does not suffer from a vanishing gradient. Contradicting the traditional way of feedforward neural networks, they used forward and introduced sideways and zig-zag interactions, which leads to better network predictions.

## 3.3 GENERATIVE ADVERSARIAL NETWORKS

The task of Generative Adversarial Networks (GANs) is to learn the distribution of the training data and generate new samples. GAN learns the distribution of the input data $\rho_{data}$ and generates synthetic images which have nearly the same distribution as the input image. The GAN architecture consists of basically two deep neural network architecture: (a): the generative network $G$ and (b): the discriminator network $D$. The generator takes a noise vector $z$ as an input from a known distribution $\rho_z$, generally a uniform distribution and outputs $G(z)$ which maps to the space of distribution $\rho_g$. The generator $G$ tries to produces more realistic images as the training progresses, and the discriminator $D$ improves its ability to discriminate between synthetic and authentic images. Finally, the generator aims to generate images that have the same distribution as the input images, i.e., $\rho_g = \rho_{data}$. This basically leads to min-max optimization problem where the discriminator network $D$ is trained to maximize $\log(D(x))$, where $D(x)$ is the output of the discriminator and the generator network is trained as to minimize $\log(1-D(G(z)))$. So, the objective function for GANs is a min-max objective function:

$$\min_{G} \max_{D} J(D,G)\& = \mathbb{E}_{x \sim p_x}[\log(D(x))] + \mathbb{E}_{z, \sim p_z}[\log(1 - D(G(Z)))]$$

Furthermore, the computation cost to train Vanilla GANs is exorbitant, and it takes a large amount of training data to learn the distribution of training data. This limits GANs and its variants to generate high fidelity images when the training data is less, and we have fewer computational resources.

To train GAN with limited data and low computational resources, Liu et al. (2021) proposed lightweight GAN, which can generate high fidelity images even when the training data is significantly less (50-100 images).

### 3.3.1 LIGHT-WEIGHT GAN

When training images are limited, creating high-fidelity synthetic images using GANs becomes a challenging task. Training GANs with limited data and inadequate computational resources can lead to over-fitting of the model and mode collapse (Arjovsky & Bottou, 2017; Zhang & Khoreva, 2018). In order to address the issue of mode collapse and overfitting while training GANs with fewer images, Liu et al. (2021) proposed a light-weight GAN method, named FastGAN, that uses skip-connections. These Skip-Layer Excitations (SLE) improves model weight gradient flow for more robust training. In a single action, SLE combines the power of style-modulation, skip-connection, and channel attention. They designed a Discriminator (D), which consists of a decoder and a feature

encoder. At each resolution in *G*, the light-weight GAN architecture utilizes a single convolution layer, including a modified skip-connection module known as the (SLE) module. As this SLE module aids in producing high-fidelity images, the requirement for high computational resources reduces. This also assists in making the model shallower, resulting in fewer parameters and requiring less computational resources, leading to faster training and smoother gradient flow.

Following is the definition of the skip-layer excitation module:

$$y = F(x_{low}, W_i) \times x_{high} \tag{1}$$

where *x* and *y* are the input and output feature maps and $\{W_i\}$ are the trainable weights.

$x_{low}$ and $x_{high}$ are two distinct dimensional inputs in SLE. $x_{low}$ is first down-sampled into (4×4) along the spatial-dimensions by an adaptive average-pooling layer in *F*, then further down-sampled into (1×1) by a convolution-layer.

## 4 MISTAKE-DRIVEN IMAGE CLASSIFICATION

We introduce a novel method for mistake-driven image classification as well as a selection of recent methodological advancements as key components in its implementation. We first present the general, high-level mistake-driven training workflow compatible with any image classification training procedure and then introduce the specific implementation and its components.

**Training Workflow**    The basic workflow of our mistake-driven image classification methodology is outlined in Figure 1 and more precisely defined in Algorithm 1. First, the initial model is trained in a standard procedure (line 1), either from scratch or in a transfer learning manner from a pre-trained backbone model. Afterward, it is validated using the validation dataset (line 2). From the validation results, the class-wise accuracy is computed (line 3) to identify the worst-performing classes (line 4). For each of these worst-performing classes, a class-specific GAN is trained on all samples in $D_{train}$ that belong to this class (line 7). Once the GAN has been trained, it is used to generate new training samples for this class (line 8), which are added to the new, augmented dataset (line 9). This step can be accelerated by parallel execution, as the individual class-specific GANs are independent of each other. Finally, the initially trained model is finetuned with the new, augmented dataset (line 11) and is available as the final model for usage (line 13) or testing (line 12).

---

**Algorithm 1** Algorithm for Mistake-Driven Training

---

**Require:** Dataset *D* with size $|D|$ and *n* classes, split into train/val/test sets;
    `nWPC`: number of worst-performing classes to augment;
    `nSamples`: number of new samples for each class;
1: Train model *m* with dataset $D_{train}$ and $D_{val}$   ▷ Either from scratch or transfer from backbone
2: Validate model *m* with dataset $D_{val}$
3: *CWA* ← Compute Class-Wise Accuracy
4: *WPC* ← Bottom `nWPC` classes in *CWA*        ▷ Select worst-performing classes (*WPC*)
5: *ND* ← $D_{train}$              ▷ Initialize new dataset from original dataset
6: **for each** class C ∈ *WPC* **do**
7:     Train class-specific GAN on $D_{train}$ data for class C
8:     *G* ← Generate `nSamples` new samples for class C
9:     *ND* ← *ND* ∪ *G*              ▷ Add generated samples to dataset
10: **end for**
11: Finetune model *m* with augmented dataset *ND*
12: Test model *m* with dataset $D_{test}$ (optional)
13: **Return** model *m*

---

The number of worst-performing classes to augment `nWPC` is a hyperparameter of the mistake-driven training procedure and balances the expansion of the dataset for finetuning versus the computational cost of the method, due to the class-wise GAN training. If `nWPC` $= n$, the method corresponds to a full GAN-based data augmentation procedure, if `nWPC` $= 0$, mistake-driven training is disabled. However, for the practical use of mistake-driven training it should be set as `nWPC` $\in [0, 0.5n]$.

The other method-specific hyperparameter is `nSamples` (line 7), which controls how many new samples are created per class. We found `nSamples` $= |C|/2$, i.e. expanding the class size by 50%, to be a suitable choice.

**Components**   The overall efficiency and final accuracy of the trained model is influenced not only by the training procedure, but, of course, especially the GAN method for the data augmentation step and the other selected components, such as the deep neural network model architecture, the backbone model for transfer learning, or the general training hyperparameters.

The GAN method is a crucial part in the training pipeline as it should be fast and data-efficient, as it is trained in a class-wise manner with only a subset of the total dataset size, and still well-performing to generate new samples that aid the finetuning process for the initially weak classes. All these characteristics are covered by the light-weight FastGAN method introduced by Liu et al. (2021) and described in Section 3.3.1. During preliminary experiments, we found the FastGAN training to be considerably faster and more data-efficient with few images than other GAN techniques, e.g., TransferGAN (Wang et al., 2018a) or StyleGAN (Karras et al., 2019), as also shown by Zhao et al. (2020), including a reduction in training time by a factor 4 from around 8 to 2 hours on the same hardware. We show a selection of generated samples during mistake-driven training in comparison to images generated by other GAN techniques in Appendix A.2.

Whereas the other components are exchangeable depending on the circumstances, we focus on the application of mistake-driven training on top of the recent methodologies from the literature to push the boundaries of the state-of-the-art in image classification. Therefore, we select Progressive SpinalNet as the key neural network architecture, as it has been shown empirically to be suitable for transfer learning and finetuning tasks (Kabir et al., 2020; Chopra, 2021), combined with Sharpness-Aware Minimization (Foret et al., 2020) for model optimization. As part of our experimental evaluation, we further consider alternative choices to evaluate the general ability of mistake-driven training to boost neural network models by only considering the weakest classes.

## 5 EXPERIMENTATION EVALUATION

We perform an extensive experimental evaluation of the proposed mistake-driven training methodology. To evaluate the contribution of each individual component in the methodology, we perform ablation studies where we remove or replace the component with alternatives.

### 5.1 EXPERIMENTAL SETUP

We perform experiments on five widely known datasets: CIFAR-10 (Krizhevsky et al., 2014), Caltech-101 (Fei-Fei et al., 2004), Stanford Cars (Krause et al., 2013), and Architectural Heritage (Llamas et al., 2017). A description of each dataset is given in Appendix **??**. For datasets with more than 20 classes, i.e. Caltech-101 and Cars, we set the number of worst-performing classes to select `nWPC` $= \lceil 0.2\,n \rceil$ to limit the computational cost of the method, otherwise, i.e. for CIFAR-10, HAM10000, and Heritage, we set `nWPC` $= \lceil 0.5\,n \rceil$. The number of newly generated samples per class is always `nSamples` $= \lceil 0.5\,|C| \rceil$, i.e. 50% of the class size rounded towards the next-biggest integer.

We train the model, both initially and during finetuning, for 20 epochs, except for CIFAR-10 with 30 epochs, and with a batch size of 28 for Caltech-101, HAM10000, and CIFAR-10, respectively 20 for Cars and Heritage. All other hyperparameters were kept to their default values as proposed by their respective authors. There was no additional preprocessing of the images performed. We report the accuracy averaged over three runs with different random seeds.

All of our experiments were performed on a Nvidia Geforce RTX 2080 Ti GPU. We used the Pytorch and Keras frameworks for our experiments.

### 5.2 EXPERIMENTAL RESULTS

As the main experiment, we evaluate our main configuration, consisting of progressive SpinalNet with a Wide-ResNet-101 (Zagoruyko & Komodakis, 2016) backbone pre-trained on ImageNet and SELU activation function, and sharpness-aware minimization. We compare both against the same

configuration without mistake-driven training, i.e., only the initial training on the dataset and the state-of-the-art accuracy as reported in the literature. The results are shown in Table 1, the best-reported accuracies over all experiments, including the ablation experiments, are marked in bold. Our main configuration reaches new state-of-the-art accuracy on Caltech-101, Stanford Cars, and HAM10000. On the architectural heritage dataset, we also reach a new state-of-the-art accuracy of 96.50%, although with a different configuration (see Section 5.3 below).

Table 1: Results for our main configuration of mistake-driven image classification with progressive SpinalNet, Wide-ResNet backbone and sharpness-aware minimization (WRN+SELU+SAM). State-of-the-art results are given as reported in the literature.

| Configuration | CIFAR-10 | Caltech-101 | Cars | Heritage | HAM10000 |
|---|---|---|---|---|---|
| Ours | 98.92 | **98.40** | **96.60** | 95.23 | **95.20** |
| w/o Mistake-Driven Training | 98.65 | 98.05 | 95.95 | 95.58 | 94.78 |
| State-of-the-art | **99.70** | 97.76 | 96.32 | 95.57 | 93.40 |

For experiments performed on the *CIFAR-10* dataset, the SOTA accuracy has been reported by Foret et al. (2020), in which the authors used EfficientNet L2 (Tan & Le, 2019) as a backbone for training the model with SAM as an optimizer. These results were not matched with our training setup, but we still observe an improvement in the final accuracy from mistake-driven training. One reason why our method might be less effective for CIFAR-10 lies in the comparatively small 32x32 image size, which constrains the expressiveness and effectiveness of the GAN method to generate new and meaningful training data, compared to the other datasets. Still, CIFAR-10 serves as a good indicator for the limitations of our methodology and gives a guideline for possible application areas. All other datasets used in our experiments have higher-resolution images, which helps the GANs to generate good-quality images.

For *Caltech-101*, the previous state-of-the-art accuracy was reported by Chopra (2021). Here we achieve an improvement of 0.64% using the same model architecture. One part of the improvement (0.29%) stems from using sharpness-aware minimization, and the other 0.35% originates from mistake-driven training. For the *Stanford Cars* dataset, mistake-driven training boosts the accuracy by 0.65% and exceeds the state-of-the-art from Ridnik et al. (2021) by 0.28%. On the *architectural heritage* dataset, our main configuration has an accuracy gap of 0.34%, but a configuration of Wide-ResNet, GELU activation function and Adam optimizer exceeds the SOTA of 95.57% from Abed et al. (2020) by 0.93%, i.e. 96.50%, as we observe in the ablation experiments. Finally, for the medical image dataset *HAM10000*, already the main configuration reaches new SOTA (94.78%), compared to the previous result by Datta et al. (2021) (93.40%), but mistake-driven training further improves the final result to 95.20%.

In conclusion, we identify improvements in final model accuracy from mistake-driven training that can reach new state-of-the-art accuracies on several datasets. Parts of the improvement result from recent advancements in model architectures, i.e., progressive SpinalNet, and model optimization, i.e., SAM, but we observe a clear benefit from mistake-driven training and focusing on the worst-performing classes rather than performing a full-sized GAN-based data augmentation procedure.

## 5.3 ABLATION EXPERIMENTS

We identified the general benefit and effectiveness of mistake-driven training for image classification. To better understand the effects of the components in the training methodology and its setup, we perform a series of ablation experiments where we remove or replace individual components. For each of the experiments, we present the results with and without mistake-driven training involving GAN augmentation. In the following discussion, we show only the results for the individual ablation experiments. Again, we highlight in bold the best results overall experiments reported in this paper. The detailed results for all different configurations considered in the experiments are shown comprehensively in Table 7 in Appendix A.3.

**Number of Worst-Performing Classes** The number of worst-performing classes were varied for the Architectural Heritage dataset (10 classes) with the configuration WRN + GELU + Adam. We

see that the number of worst-performing classes has an influence on the final performance, although there is a point of saturation where augmenting additional classes does not further improve the final performance. However, in any case the mistake-driven augmentation advances the accuracy over using no augmentation.

Table 2: Experiments performed for number of worst performing classes: We changed the nWPC to see its effect on the final accuracy.

| Heritage Dataset (10-class) | 2-class | 4-class | 5-class | 6-class | 7-class | w/o |
|---|---|---|---|---|---|---|
| WRN + GELU + Adam | 96.02 | 96029 | 96.50 | 96.50 | 96.30 | 95.22 |

**Number of Samples to Generate**   Next, the number of samples to be generated per selected class were varied, again using the Architectural Heritage dataset and the configuration WRN + GELU + Adam. We observe that a minimum number of new samples is necessary to have a relevant impact on the final accuracy, as it is to be expected. However, we also notice that when adding more than 50% new samples, we introduce an imbalance in the dataset and the total accuracy and especially the accuracy of the non-augmented classes decreases.

Table 3: Experiments performed for number of samples to generate: We changed the generated samples to see its effect on the final accuracy.

| Heritage Dataset (10-class) | 50% | 25% | 10% | No Augmentation |
|---|---|---|---|---|
| WRN + GELU + Adam | 96.50 | 96.01 | 95.94 | 95.22 |

**Backbone Models**   We vary the choice of the pre-trained SpinalNet backbone model. Similar to the experiments in Kabir et al. (2020) and Chopra (2021), we select Wide-ResNet-101 (WRN) and EfficientNet-B7 (ENB7) (Tan & Le, 2019) pre-trained on ImageNet as backbone models. We compare the configurations *WRN+SELU+Adam* and *ENB7+SELU+Adam* and show the results in Table 4. A moderate boost from mistake-driven training is observed for both configurations, i.e., the applicability of mistake-driven training is independent of the chosen model. The accuracy itself is comparable for some datasets, i.e., CIFAR-10, Cars, and HAM10000, but for others, Wide-ResNet clearly outperforms the EfficientNet model, i.e., Caltech-101 and Architectural Heritage.

Table 4: Backbone Models: We exchange Wide-ResNet-101 for EfficientNet-B7. Abbreviations: WRN - Wide-ResNet, ENB7 - EfficientNet B7

| Configuration | CIFAR-10 | Caltech-101 | Cars | Heritage | HAM10000 |
|---|---|---|---|---|---|
| WRN + SELU + Adam | 98.56 | 98.10 | 88.85 | 95.65 | 93.81 |
| w/o Mistake-Driven Training | 98.45 | 97.52 | 88.46 | 94.72 | 93.50 |
| ENB7 + SELU + Adam | 98.56 | 95.23 | 88.83 | 94.15 | 93.42 |
| w/o Mistake-Driven Training | 98.23 | 94.63 | 87.97 | 92.10 | 93.17 |

**Optimizer**   Our full setup includes Sharpness-Aware Minimization (SAM) to minimize the loss during training. As an alternative baseline, we choose the Adam training technique (Kingma & Ba, 2014) instead. The results are shown in Table 5. Again, mistake-driven training and GAN-based augmentation of the worst-performing classes boost the final accuracy in all cases, except for the Architectural Heritage dataset. Here, we observe a slight decrease of 0.35% in accuracy from mistake-driven training in combination with SAM. We also observe the benefit of using SAM, which by itself leads to better performance on all datasets compared to standard Adam. This is especially visible for the Stanford Cars dataset, where using SAM improves the accuracy by 7.49% respectively 7.75% when mistake-driven training is also used.

Table 5: Optimizer: We exchange sharpness-aware minimization for standard Adam.

| Configuration | CIFAR-10 | Caltech-101 | Cars | Heritage | HAM10000 |
|---|---|---|---|---|---|
| WRN + SELU + SAM | 98.92 | **98.40** | **96.60** | 95.23 | **95.20** |
| w/o Mistake-Driven Training | 98.65 | 98.05 | 95.95 | 95.58 | 94.78 |
| WRN + SELU + Adam | 98.56 | 98.10 | 88.85 | 95.65 | 93.81 |
| w/o Mistake-Driven Training | 98.45 | 97.52 | 88.46 | 94.72 | 93.50 |

**Activation Functions**   We further consider the choice of the activation function in the trained network and its effect on the final accuracy. Besides the main choice of the Scaled Exponential Linear Unit (SELU) activation function (Klambauer et al.), we alternatively consider the Gaussian Error Linear Unit (GELU) (Hendrycks & Gimpel, 2020) and Rectified Linear Unit (RELU) activation functions:

$$\text{SELU}(x) = \lambda \begin{cases} x & \text{if } x > 0 \\ \alpha e^x \text{-} \alpha & \text{if } x \leq 0 \end{cases}$$

$$\text{GELU}(x) = x \cdot \frac{1}{2} \left( 1 + \text{erf} \left( \frac{x}{\sqrt{2}} \right) \right)$$

$$\text{RELU}(x) = max(x, 0)$$

We show the results in Table 6. Similar to the previous experiments, mistake-driven training consistently boosts the final accuracy. Additionally, in the configuration of Wide-ResNet (WRN), GELU, and Adam, the model reaches a new state-of-the-art accuracy on the architectural heritage dataset with 96.50% accuracy compared to the previous 95.57% reported by Abed et al. (2020).

While our main configuration without mistake-driven training achieves a higher accuracy than *WRN+GELU+Adam* without mistake-driven training and already matches the SOTA (95.58% vs. 95.57%), we observe no boost from mistake-driven training in the main configuration, whereas it is observable for the changed configuration and leads to a further improvement in accuracy.

Table 6: Optimizer: We exchange the SELU activation function for GELU and RELU.

| Configuration | CIFAR-10 | Caltech-101 | Cars | Heritage | HAM10000 |
|---|---|---|---|---|---|
| WRN + SELU + Adam | 98.56 | 98.10 | 88.85 | 95.65 | 93.81 |
| w/o Mistake-Driven Training | 98.45 | 97.52 | 88.46 | 94.72 | 93.50 |
| WRN + GELU + Adam | 98.20 | 97.50 | 89.44 | **96.50** | 94.25 |
| w/o Mistake-Driven Training | 97.92 | 97.22 | 88.46 | 95.22 | 94.03 |
| WRN + RELU + Adam | 98.70 | 97.45 | 91.16 | 95.65 | 93.63 |
| w/o Mistake-Driven Training | 98.19 | 97.28 | 88.83 | 95.37 | 93.28 |
| State-of-the-art | **99.70** | 97.76 | 96.32 | 95.57 | 93.40 |

## 6   CONCLUSION

This paper proposes a novel, mistake-driven training technique for image classification. Our proposed method uses pre-trained models as a base for training and, after initial training, extracts classes with high losses for class-wise augmentation and solves the problem of imbalanced class performance. A challenge in this process is the high resolution of images that we consider as datasets in combination with a limited data regime. Therefore, we rely on the light-weight FastGAN method for GAN-based data augmentation, which is both data-efficient and fast and thereby highly suitable for our method. We have also implemented the sharpness-aware minimization optimizer, and the results reveal that we obtain the highest performance when using the SAM optimizer in most cases. In our experiments, except for the CIFAR-10 dataset, our proposed training method outperforms all the previous SOTA accuracies. While we have focused on image classification in this work, we plan to generalize the method to other classification tasks in the future.

**Reproducibility Statement** The presented method and experiments are based upon freely available and widely used methods and datasets, enabling the reproduction of the method and the reported results. We will also release the source code for the experiments once the paper is accepted.

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

## A APPENDIX

### A.1 DATASETS

**CIFAR-10** The CIFAR-10 dataset (Krizhevsky et al., 2014) contains 60,000 colored images of size 32x32 pixels divided into ten classes with each 6,000 images. The dataset is split 50,000/10,000 into training and test set.

**Caltech-101** This dataset (Fei-Fei et al., 2004) is the most imbalanced dataset with 101 different classes. This dataset has colored images of size 300x200 pixels. The number of images in each class varies from 40 to 800. However, on average, each class has about 50 images.

Note: Due to the imbalance between classes in this dataset, after initial training, out of 101, we extracted the bottom 20 classes with the most misclassification rate and used only those classes for our final experiments. This is done assuming that if our model gives good results on the worst-performing classes, it will give high-quality results on the well-performing classes.

**Stanford Cars** The Stanford cars dataset (Krause et al., 2013) consists of 16,185 colored images representing 196 different classes. The data is divided into 8,144 training photos and 8,041 testing images, with about a 50-50 distribution between training and testing for each class.

**Architectural Heritage** There are three types of architectural heritage elements datasets (Llamas et al., 2017). One with 64x64 image size, second with 128x128 image size, and third with 224x224 image size. In our experiments, we have used the second dataset with an image size of 128x128. The dataset consists of 10,235 images for all ten classes.

**HAM10000 Skin Lesions**   HAM dataset (Tschandl, 2018) stands for humans against a machine which consists of 10,000 dermatoscopic training images. It consisted of colored images of 1872x1053 pixels, which were then manually cropped to 800x600 pixels centering the lesion.

## A.2   EXEMPLARY GENERATED IMAGES

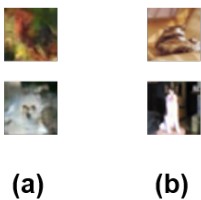

(a)          (b)

Figure 2: CIFAR-10: A comparison between original and generated images for CIFAR-10 dataset; (a) Original images, (b) Generated images using FastGANs.

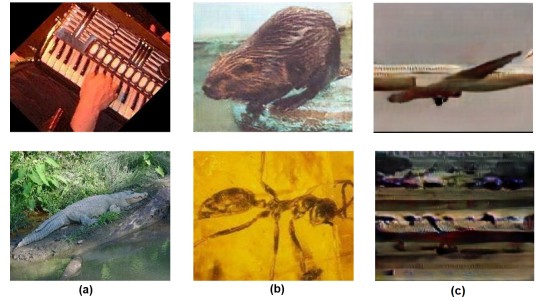

(a)          (b)          (c)

Figure 3: Caltech-101: A comparison between original and generated images for Caltech-101 dataset; (a) Original images, (b) Generated images using FastGANs. (c) Generated images using pretrained GANs (Zhao et al., 2020).

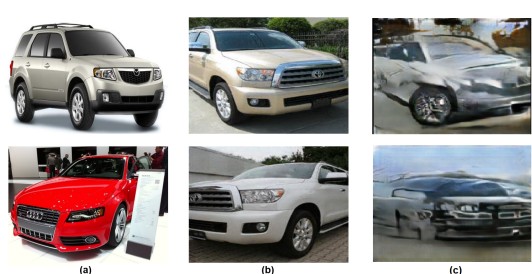

(a)          (b)          (c)

Figure 4: Cars: A comparison between original and generated images for Stanford Cars dataset; (a) Original images, (b) Generated images using FastGANs. (c) Generated images using pretrained GANs (Zhao et al., 2020).

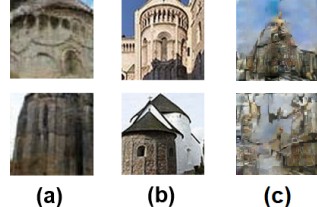

**(a)**   **(b)**   **(c)**

Figure 5: Heritage: A comparison between original and generated images for Architectural Heritage dataset; (a) Original images, (b) Generated images using FastGANs. (c) Generated images using pretrained GANs (Zhao et al., 2020).

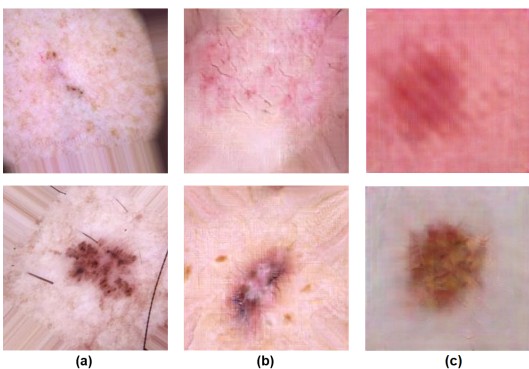

(a)   (b)   (c)

Figure 6: HAM10000: A comparison between original and generated images for Skin Cancer MNIST: HAM-10000 dataset; (a) Original images, (b) Generated images using FastGANs. (c) Generated images using pretrained GANs (Zhao et al., 2020).

## A.3    DETAILED RESULTS FOR EXPERIMENTS

Table 7: Experimental results of mistake-driven image classification in different configurations. Abbreviations: WRN - Wide-ResNet, ENB7 - EfficientNet B7

| Configuration | CIFAR-10 | Caltech-101 | Cars | Heritage | HAM10000 |
|---|---|---|---|---|---|
| WRN + SELU + SAM | 98.92 | **98.40** | **96.60** | 95.23 | **95.20** |
| w/o Mistake-Driven Training | 98.65 | 98.05 | 95.95 | 95.58 | 94.78 |
| WRN + SELU + Adam | 98.56 | 98.10 | 88.85 | 95.65 | 93.81 |
| w/o Mistake-Driven Training | 98.45 | 97.52 | 88.46 | 94.72 | 93.50 |
| WRN + GELU + Adam | 98.20 | 97.50 | 89.44 | **96.50** | 94.25 |
| w/o Mistake-Driven Training | 97.92 | 97.22 | 88.46 | 95.22 | 94.03 |
| WRN + RELU + Adam | 98.70 | 97.45 | 91.16 | 95.65 | 93.63 |
| w/o Mistake-Driven Training | 98.19 | 97.28 | 88.83 | 95.37 | 93.28 |
| ENB7 + SELU + Adam | 98.56 | 95.23 | 88.83 | 94.15 | 93.42 |
| w/o Mistake-Driven Training | 98.23 | 94.63 | 87.97 | 92.10 | 93.17 |
| State-of-the-art | **99.70** | 97.76 | 96.32 | 95.57 | 93.40 |

