# OpenReview forum: "Mistake-driven Image Classification with FastGAN and SpinalNet"
_ICLR.cc/2022/Conference — ICLR 2022 Submitted_

### Official Review · Reviewer_nk5o · 2021-10-28

**Correctness:** 3
**Technical Novelty And Significance:** 2
**Empirical Novelty And Significance:** 2
**Recommendation:** 1
**Confidence:** 5

**Main Review:**

Strengths:
- The paper proposes to engineer a way to augment low-accuracy class data through a recent lightweight GAN model.

Weaknesses:
- The novelty is limited, the combination of a pertained classifier and its output to evaluate low accuracy classes is followed by separate GAN generators for each of those classes.
- Five relatively old datasets are used to evaluate the method. However, in the experimental results, it is not clear which and how many classes in those datasets were selected by the model as the lower accuracy classes, hence were augmented into the dataset, and how that affected the performance.
- Also, it is not possible to judge the quality of the FastGAN-based  augmentations for the imbalanced (or low-accuracy?) classes, as they were not provided.


**Summary Of The Paper:**

The paper presents a method that uses a pertained model for classification in order to choose classes with low accuracy, then performs a GAN-based augmentation to those classes, and retrains the classification model with the augmented dataset. The claim is that this procedure improves the classification performance. There are several known components in the proposed architecture, and recent works such as SpinalNet, and FastGAN are utilized as a classifier and as a lightweight GAN synthesizer, respectively.

**Summary Of The Review:**

My opinion is that the idea in the paper is at best a nice engineering idea, which aims to improve on the class imbalance problem, which typically lead to lower accuracy for those classes.

The novelty of the paper is limited, due to reasons listed above.  ICLR is not the appropriate venue to present this idea.

---

> ### Author Response · Authors · 2021-11-16
> **Response to Reviewer**
>
> We thank the reviewer for their effort in reviewing the paper and their comments.
>
> > Five relatively old datasets are used to evaluate the method
>
> We politely disagree with the reviewer’s comment on the datasets being (too) old. Some of the datasets we used were published in 2017 (Heritage Dataset) and updated in July 2021. Ham skin lesson dataset was published in 2018 and the Stanford cars dataset was published in 2013. The main reason behind using these datasets is to show that our method is generic enough to be used on a variety of datasets that are already known in the literature.
>
> > in the experimental results, it is not clear which and how many classes in those datasets were selected by the model as the lower accuracy classes
>
> We would like to kindly refer to the description in the Experimental Setup (Sec. 5.2):
> For datasets with more than 20 classes, i.e. Caltech-101 and Cars, we set the number of worst-performing classes to select $\texttt{nWPC} = \lceil0.2\,n\rceil$ to limit the computational cost of the method, otherwise, i.e. for CIFAR-10, HAM10000, and Heritage, we set $\texttt{nWPC} = \lceil0.5\,n\rceil$. The number of newly generated samples per class is always $\texttt{nSamples} = \lceil 0.5\,|C| \rceil$, i.e. 50\% of the class size rounded towards the next-biggest integer.
> The next revision of the manuscript will present the selected values more clearly.
>
> Also, we have performed additional experiments to better understand the setting of these variables. Please see the answer to reviewer 2.
>
> > Also, it is not possible to judge the quality of the FastGAN-based augmentations for the imbalanced (or low-accuracy?) classes, as they were not provided.
>
> We have provided samples of generated images in the appendix, additionally the quality in terms of performance improvement is visible through the ablation studies performed in the paper. It is important to note that the generated images are never actually presented to a human or should follow the standard of high-fidelity image generation. Their purpose is to support the training process only.

---

### Official Review · Reviewer_N1Vg · 2021-11-02

**Correctness:** 3
**Technical Novelty And Significance:** 1
**Empirical Novelty And Significance:** 2
**Recommendation:** 3
**Confidence:** 5

**Main Review:**

Strength:
1. The proposed method make sense and boosts the performance.
2. The paper is clearly written and easy to repeat.

Weakness:
1. The proposed method in this paper can be regarded as an engineering trick on training procedure of CNN models. The adopted algorithms are existing ones or based on well-studied methods. This makes the method show limited technical novelty.
2. The performance enhancements are quite marginal on those tested datasets. For most of cases, the improvement is less than 1%, which is not impressive, considering to cost of generating extra images and extra fine-tuning step.
3. The experiments are not convincing. The baseline algorithms have already achieved good performance on five tested datasets, e.g., mostly higher than 95%. It is more convincing to conduct experiments on more challenging datasets and datasets with larger-scale, e.g., the Imagenet.
4. This paper also lacks necessary ablation studies on some important parameters, like the nWPC.


**Summary Of The Paper:**

This paper studies mistake-driven image classification, with the idea of applying data augumentation to classes with lower classification accuracy. Emphasizing the training on classes with low accuracy boosts the overall performance of trained CNN classifier.

The proposed method is quite straightforward, and the paper is clearly written. It first evaluates the classification accuracy of each class on the validation set, then selects the categories with lowest accracy for data augumentation with GAN, and add the augumented class into the training set for fine-tuning. Experiments are tested on 5 dataset, which demonstrate improved performance.

**Summary Of The Review:**

To summarize, this paper is clearly written, and demonstrates an engineering trick for training cnn classifiers. It boosts the classification accuracy at costs of larger computation and memory consumptions. The experiments are also not convincing enough.

---

> ### Author Response · Authors · 2021-11-16
> **Response to Reviewer**
>
> ## Response to Comments
>
> We thank the reviewer for their effort in reviewing the paper and their comments.
>
> > The proposed method in this paper can be regarded as an engineering trick
>
>
>
> > The performance enhancements are quite marginal on those tested datasets
> > The baseline algorithms have already achieved good performance on five tested dataset
>
> Our main task is to create a reliable model which regardless of the type of dataset gives the best results.
> We would like to highlight that especially for already high baseline accuracies the improvement by 1% is a substantial improvement, in our case leading to state-of-the-art accuracies on these datasets.
> We agree that a small improvement can be seen as marginal if the initial accuracy is very low, but in the competitive setting of the chosen baseline methods already a small improvement is relevant.
>
> > This paper also lacks necessary ablation studies on some important parameters, like the nWPC.
>
> We have presented a number of ablation studies on the different components in the method already. We have performed additional experiments on the number of worst-performing classes (`nWPC`) and number of samples to be generated (`nSamples`). For these parameters we give practical recommendations for their values in the paper already.
>
> ## Number of worst-performing classes
>
> We have varied the number of worst-performing classes for the Architectural Heritage dataset (10 classes) with the configuration WRN + GELU + Adam.
>
> 1. 20% (2 classes) = 96.02%
> 2. 40% (4 classes) = 96.29%
> 3. 50% (5 classes) = 96.50% (used throughout the experiments)
> 4. 60% (6 classes) = 96.50%
> 5. 70% (7 classes) = 96.30%
> 6. no augmentation = 95.22%
>
> We see that the number of worst-performing classes has an influence on the final performance, although there is a point of saturation where augmenting additional classes does not further improve the final performance. However, in any case the mistake-driven augmentation advances the accuracy over using no augmentation.
>
> ## Number of samples to be generated
>
> We have also varied the number of samples to be generated per selected class, again using the Architectural Heritage dataset and the same configuration.
>
> 1. 50% new samples = 96.50% (used throughout the experiments)
> 2. 25% new samples = 96.01%
> 3. 10% new samples = 95.94%
> 4. No augmentation = 95.22%
>
> We see that a minimum number of new samples is necessary to have a relevant impact on the final accuracy, as it is to be expected. However, we also notice that when adding more than 50% new samples, we introduce an imbalance in the dataset and the total accuracy and especially the accuracy of the non-augmented classes decreases.
>
> We will revise the paper manuscript to include these results.

---

### Official Review · Reviewer_hoDS · 2021-11-04

**Correctness:** 4
**Technical Novelty And Significance:** 1
**Empirical Novelty And Significance:** 1
**Recommendation:** 3
**Confidence:** 4

**Main Review:**

-The paper tries to adress an ongoing challenge in machine learning.   The idea is simple and efficiently works. However, I do not believe the method is novel. For example, for incremental class tasks with no samples from the previous classes, GANs are widely used to generate the new samples for those classes. Or even the GAN is widely used for augmentation. It seems this paper just exploited that method for a new task.

Besides, as I understand, the low accuracy will be associated with the class with a limited number of samples. How do you train a GAN on such tiny samples?

Also, there are several methods for dealing with the imbalanced training set, for example, downsampling the small classes or upsampling the large classes. Did this method work better? If so, please put the result on the paper.


**Summary Of The Paper:**

The paper proposes a solution for the imbalanced classification method.  Learning the CNN traditionally on training data, Finding the classes with low accuracy and generating/augmenting the samples from those classes aiming to improve the accuracy of those classes by fine-tuning the model on it, is the proposed solution.  Several experiments on datasets such as CIFAR10 is done. Results the feasibility and perfection of the idea.  For generating samples from classes with low accuracy,  a GAN is used.

**Summary Of The Review:**

-The paper is not novel. The idea is previously exploited for other tasks.
-Experiments is not completed.

---

> ### Author Response · Authors · 2021-11-16
> **Response to Reviewer**
>
> We thank the reviewer for their effort in reviewing the paper and their comments.
>
> > I do not believe the method is novel. For example, for incremental class tasks with no samples from the previous classes, GANs are widely used to generate the new samples for those classes. Or even the GAN is widely used for augmentation. It seems this paper just exploited that method for a new task.
>
> > there are several methods for dealing with the imbalanced training set
>
> As the reviewer mentioned, our paper does not propose a solution for the imbalanced _dataset_ problem. Our paper proposes a method that focuses on imbalanced _performance_ in image classification, where the trained model has performance deficits in some of the dataset’s classes after an initial training phase.
> We agree with the reviewer that GANs are widely used for augmentation. However, we do not claim our contribution in augmenting data. Augmentation of data is one part of the whole method.
>
> By combining and extending existing methodology, we are able to transfer towards new tasks and can achieve state-of-the-art results with a simple, but straightforward method.
>
> > the low accuracy will be associated with the class with a limited number of samples. How do you train a GAN on such tiny samples?
>
> We want to clarify that our method is not focused on imbalanced datasets. The augmented classes are the _worst-performing classes_ after the initial training. For each of these classes we use all of their training samples for the GAN training.
> It is correct that this subset might be comparatively small to large-scale GAN applications, but the recent developments in FastGAN (a light-weight GAN method, see Section 3.3.1) make it feasible to train GAN models for sufficiently high quality with limited data. Also keep in mind that the actually generated images need to be sufficient for the improved training of the classification model, but not necessarily to the same standard as high-fidelity GAN generation.
>
> > there are several methods for dealing with the imbalanced training set, for example, downsampling the small classes or upsampling the large classes. Did this method work better?
>
> As mentioned, the focus is on imbalanced performance, not imbalanced datasets. We have therefore not further considered methods specifically designed for that use case.

---

### Decision · Program_Chairs · 2022-01-20

**Decision:**

Reject

**Comment:**

The paper received 3,3,1 as reviews. All reviewers have the consensus on the weaknesses, i.e. limited technical novelty and weak boost in performance in datasets that may not be the state of the art anymore. The authors have submitted a rebuttal however the rebuttal did not improve the score of the reviewers. Following the reviewers recommendation, the AC recommends rejection.